# Using Lean to Improve Operational Performance in a Retail Store and E-Commerce Service: A Portuguese Case Study

**Pedro Alexandre Marques** [1,*], **Diana Jorge** [1] **and João Reis** [1,2]

1    EIGeS—Research Centre in Industrial Engineering, Management and Sustainability, Lusófona University, Campo Grande, 376, 1749-024 Lisbon, Portugal; diana.jorge@ulusofona.pt (D.J.); joao.reis@ulusofona.pt (J.R.)
2    Department of Military Sciences, Portuguese Military Academy and CINAMIL, Rua Gomes de Freire 203, 1169-203 Lisbon, Portugal
*    Correspondence: p5037@ulusofona.pt; Tel.: +351-21-751-5500

**Abstract:** Ensuring on-shelf availability is essential for retailers to maintain high service levels for both in-store and E-Commerce consumers. The performance of this indicator largely depends on reorder planning decisions, as well as on the effectiveness of the replenishment process. This paper presents a case study that involved two Lean initiatives, which together have contributed to a significantly reduction in the number of out-of-stock events incurred by a retail store and an increase in the order fulfilment rate accomplished by the online commerce service. In the first initiative, a value stream management (VSM) methodology was adopted to redesign the existing replenishment process in the most relevant fresh food market: fruits and vegetables. The second initiative involved the implementation of a simple, but effective visual inventory management system in the warehouse of the E-Commerce division, where a wide set of fast-moving consumer goods (FMCG) is stored using kanban cards. This paper hence demonstrates, through practical application, that Lean tools can be employed to improve operational processes with positive impacts on both the physical store performance and on results regarding the online commerce business.

**Keywords:** E-Commerce; lean management; order fulfilment rate; out-of-stock; value stream mapping





## 1. Introduction

Out-of-stock (OOS) is a major problem in the retail business, since it contributes to lost sales and decreased consumer loyalty [1–3]. Poor in-store replenishment significantly contributes to a decrease of on-shelf availability (OSA) performance, hence negatively impacting consumer behavior and sales [4,5]. Furthermore, customer order fulfilment in online retailing largely depends on the OSA performance of the physical store [6,7]. According to recent reports, the number of online purchases is expected to keep rising significantly in the coming years [8], especially after the pandemic situation caused by COVID-19 that has contributed to an acceleration in digital purchases and to the increased importance of E-Commerce in the retail business [9,10]. E-commerce is nowadays a critical part of a retailer's multi- and omni-channel strategy [11,12], being relevant to an understanding of how effective channel integration is and how it can be improved [13,14]. Due to the rapid growth of online shopping during recent years, the opportunity for the expansion of the fresh food E-Commerce market has become a reality [15]; however, the unique and intrinsic characteristics of fresh products may lead consumers to retain a preference for the supermarket as their main channel of purchasing, since virtual purchasing limits the ability to examine the purchased items [16].

Among all the areas of a food retail store, fresh food markets are of great importance, as they account for 30 percent of overall sales (usually with higher margins than packaged items) and represent around 50% of the total inventory turnover [17]. Furthermore, they have higher-than-average daily sales per item [18] and, according to Buck and Minvielle [19], they are amongst the strongest drivers of customer loyalty and store

traffic. Fresh food refers to product categories such as fruits and vegetables, meat, sea food, baked goods, among other perishable items. Due to its distinctively short shelf life, fresh food is complex to manage and particularly vulnerable to operational and supply chain risks [20]. Moreover, some fresh product categories are often subjected to specific temperature and handling requirements, leading to additional challenges when performing the replenishment process [14,21].

In order to enhance the effectiveness and productivity of their in-store, online and/or supply chain operations, many retailers have adopted operational excellence programs based on Lean principles, methods and tools to improve performance, productivity and customer satisfaction [22–26]. They seek the systematic adoption of proven methods and tools that will allow them to continuously identify and eliminate waste factors from their value streams or processes [27]. The origins of Lean can be found on the shop floors of Japanese manufacturers, in particular at the Toyota Motor Corporation [28]. However, over the past two decades, the world has embraced Lean management thinking, while a large amount of literature suggesting the benefits of Lean has accumulated [29]. Furthermore, it has spread to all kinds of industries and application areas [30], including in retail [24,31–33].

This paper presents a successful continuous improvement project conducted in 2018 at a hypermarket store of a large multinational retail enterprise operating in Portugal. In the backroom of this hypermarket there is a warehouse, managed by the E-Commerce division that is used to store a set of fast-moving consumer goods, namely the 600 best-selling product references and some high-volume items. The remaining items ordered online are picked up at the store, including fresh food items. This project aimed at increasing the order fulfilment rate achieved by the online business, which depends on the out-of-stock (OOS) rates in both the hypermarket store and the online warehouse. To reduce the OOS rate in the warehouse, a simple, but effective visual inventory management procedure was introduced through the use of kanban cards. A decrease of the OOS rate focused on the fruits and vegetables section, the most relevant fresh food market, by redesigning the existing replenishment procedure using a value stream management (VSM) methodology. Kanban and VSM are two relevant approaches available from the Lean toolbox, and in this paper a set of new VSM icons, that were specifically designed to suit the in-store operational processes of commerce and retail, were introduced for the first time. By managing inventory more effectively, as well as by ensuring greater efficiency in the replenishment of fresh products, such as fruits and vegetables, the company was also able to reduce the negative impact of fresh food waste, thus contributing to an enhancement in its performance from a sustainability point of view.

This paper also aims to demonstrate the applicability and advantages of employing Lean concepts, methods and tools in a company from the retail sector. As far as the authors are aware of, this is the first published case study describing an improvement project in the Portuguese retail sector. Another contribution of the paper is the fact that it describes how Lean can be used simultaneously to improve the performance of in-store operations and processes from the E-Commerce business. Finally, it intends to explore how a kanban pull system can contribute to a reduction in stockouts in a warehouse dedicated to online commerce, hence driving an increase in the fulfilment rate of online orders. The paper is organized around four sections. In the next section a complete literature review is performed. This is followed by the presentation of the case study, which comprises the description of the methodology adopted, the scope of the problem, the conduction of diagnosis analysis, the development and testing of solutions and the discussion of the results. Finally, the main conclusions of the paper are summarized and suggestions for future research are proposed.

## 2. Literature Review

This section is divided into two subsections: the first is related to the literature review on Lean Management (VSM and kanban), while the second presents the literature on the applications of Lean Management to retail.

### 2.1. Lean Management

The concept of Lean Management has its roots in the Toyota Production System (TPS), a manufacturing approach pioneered by the Japanese engineers Taiichi Ohno and Shigeo Shingo [34–36]. The TPS working culture promotes the empowerment of teams and continuous improvement (kaizen) practices, having been developed while Toyota was facing difficulties that were jeopardizing its survival [37,38]. It is likely that the first formal documentation on TPS were the supplier manuals published by Toyota's Purchasing Administration Department (established in 1965), in order to teach suppliers about the requirements for operating a just-in-time (JIT) delivery system using the kanban concept [39,40]. For over 30 years TPS or JIT were used interchangeably to refer to Ohno's/Shingo's efficient manufacturing system [41]. The generic term "Lean Production" came into existence from the international motor vehicle program (IMVP) research at Massachusetts Institute of Technology, being first used by Krafcik [42] and popularized by Womack et al. [43] in "The Machine that Changed the World".

Widespread interest from western manufacturers did not begin until the early 1980s [28]. However, since then, many Lean practices, grounded in TPS, have disseminated beyond Toyota to other automakers and to all kinds of industries across the globe [29,30,37], including applications in non-manufacturing areas [31,35,44–46], such as in retail [22,24–26,32]. Lean has also been successfully integrated with other management practices, including Six Sigma [34,47]. Authors such as Yadav and Desai [48] and Singh and Rathi [49] provide a complete review about the application of Lean Six Sigma in a wide range of sectors and business activities.

It is possible to encounter diverse definitions for the concept of "Lean" in the published literature [29,35,36]. In a simplistic way, Lean intends to create more value for the customer and to business, while reducing waste and cost factors for everyone [28,44,50]. The prevailing published literature describes Lean as an approach to systematically identify and eliminate waste (or muda in Japanese) in the organizational processes [29]. Historically, the following seven types of waste have been identified [51]: unnecessary movement, excess of transportation, waiting, inadequate inventory levels, overproduction, overprocessing, and defects/errors/non-conformities. More recently, waste of human capital was added as an eighth type of waste [52].

Although the focus on efficient waste-free flows remains at the center of Lean systems [36], to ensure the long-term sustainability of the results of an organization, it must be regarded as a company-wide management system [29,41,50,53]. Many companies fail to sustain and deploy their Lean programs because they misuse the application of the tools and practices with the system itself and its principles [54,55]. Liker [56] identified 14 principles, clustered under four groups, and two pillars of TPS. The pillars are continuous improvement (kaizen) and respect for people. Kaizen, the first pillar, is defined as a culture of sustained improvement targeting the elimination of waste in all systems and processes of an organization [57]. The second pillar relies on respecting every single individual [55]. The 14 principles and their corresponding groups are depicted in Figure 1.

Lean offers a wide variety of tools and techniques, which can be effectively adopted by any type of organization to eliminate anything that does not create value [57].

In the context of the case study presented in this paper, two tools assume particular relevance: Value stream management (VSM) and kanban. Each of them will be discussed in the following subsections.

### 2.1.1. Value Stream Management—VSM

The concept of "value stream" is fundamental to Lean [58–60] and it refers to all the (value-added and non-value-added) activities that an organization needs to perform to design and create its products or services and deliver to its customers [61,62]. Usually, value streams cross through multiple departments within an organization [56], providing an overview of the entire work processes and how they interact [63]. A value stream consists of processes and each process consists of activities or tasks [51]. When managing a

value stream, it is possible to consider the following types of flows: physical/material flow, information flow, cash/financial flow [31,64], flow of people [59] and flow of energy [65].

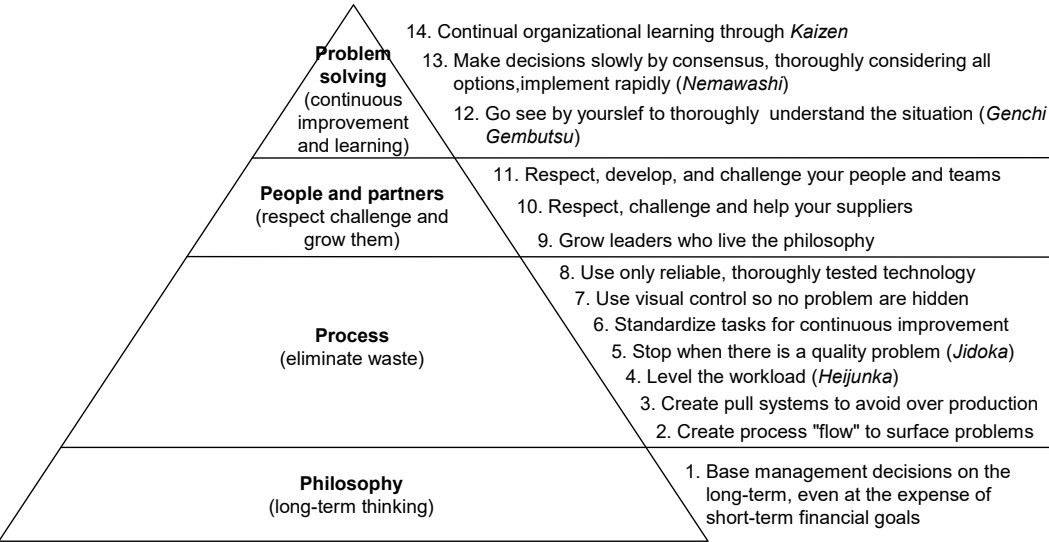

**Figure 1.** The four groups of 14 principles of TPS (adapted from: Liker [56]).

One of the first papers on the VSM issue was authored by Hines et al. [66], in which they define it as "a strategic and operational approach to the data capture, analysis, planning and implementation of effective change within the core cross-functional or cross-company processes required to achieve a truly Lean enterprise". The term "Value Stream Management" is often referred to as "Value Stream Mapping". Authors such as Erlach [67] and Oberhausen and Plapper [68], consider VSM to involve two stages: (1) Value Stream Mapping to describe and analyze the current state situation of the value stream, and (2) Value Stream Design to idealize a future state of the process.

From a tool perspective, VSM evolved from the original technique used by Toyota named "material and information flow mapping" [56]. The current version of VSM was introduced in 1999 through the book "Learning to See" by Rother and Shook [59]. This book brought a new array of tools, concepts, and graphical symbolic illustrations and icons to perform process mapping, but with the purpose of improving the visualization of process flows that encompasses a value stream, highlighting waste and areas to be improved.

The VSM methodology relies on the sequence of the following five Lean principles [31,56,69]:

1.  Specify value. Understand who the customer is, determine what their needs are and define what is "value" for the customer.
2.  Identify and analyze the value stream. Map the flow of value, perform value-added analysis and identify existing waste factors in the value stream.
3.  Make the value flow. Design, develop and implement a future state, with less waste factors, where value will flow continuously in the direction of the customer.
4.  Let the customer pull value. Standardize the value stream and initiate the flow when the customer pulls their needs.
5.  Pursue perfection. Repeat the cycle towards the creation of new future-state maps and continuously improve in order to seek perfection.

The previous sequence steps follow the well-known Plan-Do-Check-Act (PDCA) continuous improvement cycle [44,70] and it can be further detailed, as depicted in Figure 2. The choice of the value stream and its boundaries (i.e., begin and end points) is not arbitrary, since it should derive from the strategic planning and deployment decisions [67,71]. The first step regards the identification of the customers served by the selected value stream, as well as gathering the needs and desires that will determine their customer value attributes.

Then the current value stream is mapped "as-is" according to the guidelines, symbology and icons provided by the VSM tool. This visual mapping will allow a careful study of the flows presented in the value stream, thus enabling the identification of the largest wastes existing there. Based on the improvement opportunities previously identified, one can design the desired future state map for the value stream and then create and implement planned improvement actions. Quantifying the gains resulting from the improvement actions will enable the setting of conclusions regarding their effectiveness. Once the effectiveness is confirmed, all the related new processes need to be standardized.

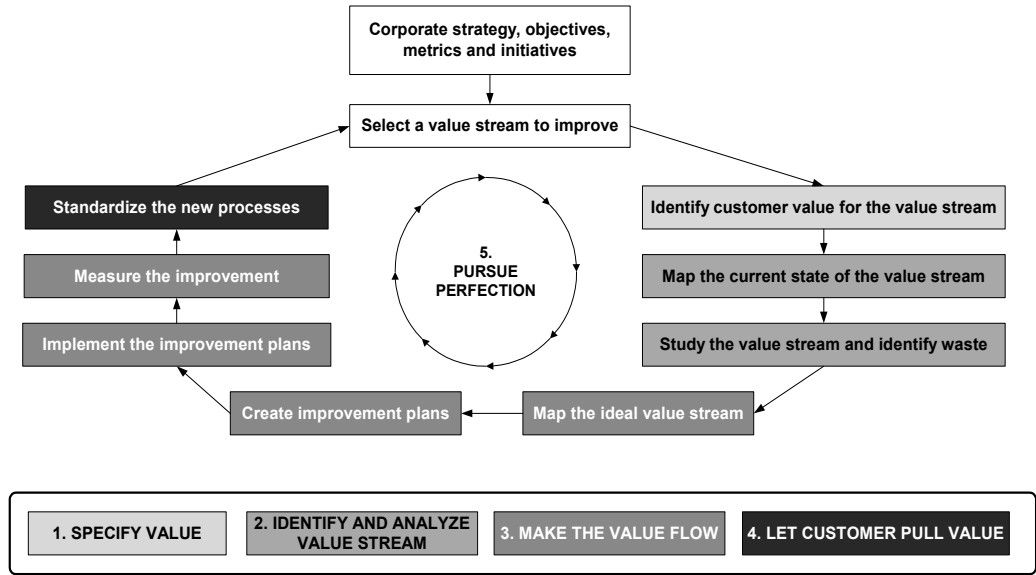

**Figure 2.** The steps of the VSM methodology and their relationship with the five Lean principles (adapted from Womack et al. [43], and Tischler [44]).

### 2.1.2. Kanban

Kanban is a subsystem of TPS that was created to control the levels of inventory to regulate production and supply of components, and in some cases, raw material [72,73]. It is the Japanese word for "visual card" or "visual signal". According to Ohno [60], the concept behind kanban was inspired by the shelf's replenishment operations of American-style supermarkets, which replaced items only when customers made a purchase. In this regard, customers get what they need, at the moment it is needed, and in the quantity needed [74,75]. Kanban was developed to signal the pulls through the value stream [28,41,76]. The kanban system provides an effective form of visual management for process control to materialize JIT production [27,74,77].

In simple terms, kanban is a signaling device that gives authorization and provides instructions for the production/ordering or withdrawal/conveyance of items in a pull system [78]. There are, however, different types of solutions to materialize a kanban pull system [41]. A kanban does not necessarily need to be a card, it can actually be a verbal command, a hand signal, a flag, a light, among other means [79], including electronic or digital solutions [80]. Kanbans can be classified according to their functions [74,75], being usual to distinguish between the single kanban card system and the dual kanban card system [41,77,81]. The latter can be further decomposed into withdrawal kanban (also known as conveyance or transportation kanban) and production kanban [72,79,82]. In addition to the previous types, Monden [83] and Braglia et al. [84] mention the signal Kanban, while Huang and Kusiak [74] grouped the different kinds of kanban together into primary kanban, supply kanban, procurement kanban, subcontract kanban and auxiliary kanban. Monden [83] also provided a list of additional types of kanban used within TPS that include express kanban, emergency kanban, through kanban, and common kanban.

In Figure 3 we summarize the categories and definitions of the different types of kanban proposed in the literature.

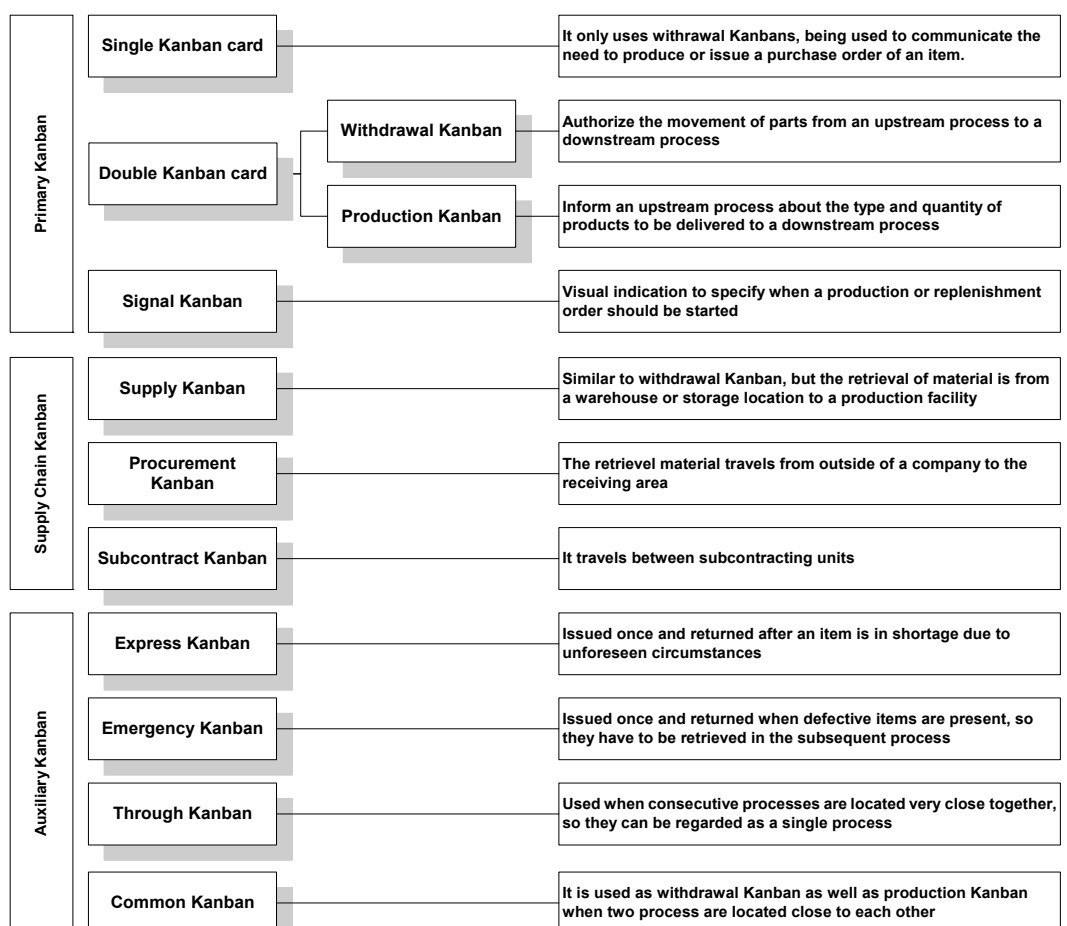

**Figure 3.** Organization of the different types of kanban referred in the literature (own study).

The usage of kanban devices fits into the information flow captured through VSM [85]. They are a key element when one is designing the future state of the value stream in order to promote a pulled and continuous flow, reduce work-in-progress inventory levels, enhance productivity, and reduce lead times [59,67,85].

### 2.2. Lean Management in the Retail Sector

Retailing comprises an important portion of the EU and US economies since it contributes to a significant portion of their gross domestic product [86]. Due to the increasing importance of E-Commerce, which grew globally at an average rate of 20% during the last decade and has accelerated during the pandemic situation, especially in the food retail sector [87], the relevance of the retail industry tends to be even greater. Given its impact on the economy, many researchers have paid attention to many of the topics surrounding retail management. Nevertheless, and despite how Lean thinking has migrated into service operations, including retail [22,88,89], relatively few publications regarding the application of these philosophies to the retail industry have been produced [89,90].

The term "Lean retailing" was introduced in the late 90s [91] and has initially become associated with Wal-Mart's early adoption of management practices and information technology systems to strengthen the relationship with its suppliers [92]. The ultimate goal of Lean retailing is to ensure a rapid and efficient flow of goods to the customers, hence ensuring a rapid replenishment of the shelves and of other points of sale [22,93,94]. To achieve this objective, waste factors in retail processes need be continuously identified

and eliminated, from in-store operations to the whole upstream processes that comprise the retail supply chain [95]. The key is therefore to manage and continuously improve the retail value streams towards the implementation of a "pull" replenishment that will enable the achievement of high levels of product availability [22,27,31,94].

Retailing is a service industry that is embracing Lean thinking [24,31,32,96], often led by well-known retail companies. Wal-Mart, the world's largest retail company, and Tesco, the leading UK retailer, have become famous examples of the adoption of Lean principles in this sector [93]. By developing closer supplier relationships and communication, as well as improving distribution and logistics processes under a "pull" vision, these two giants have been able to increase their levels of service to consumers, while reducing inventory and operational costs [24,91]. The Spanish supermarket giant Mercadona is another example of integrating Lean into its operations and processes. Its business model is organized around the customer, which the company calls "the boss" [97]. This stance is supported by the implementation of excellent operations management principles that focus on eliminating everything that does not add value to the consumer [96]. In addition to well-designed processes, well-treated and well-trained people in all core operational processes are key elements of Mercadona's model [98]. Naruo and Toma [32] studied how Lean principles and concepts were successfully applied in Seven-Eleven Japan and describes the outcome of some projects that contributed to an increase in the integration of processes from ordering to delivery, enhanced service levels, boosted sales and reduced inventory levels. Onetto [99] outlined how Lean continuous improvement (Kaizen) practices adopted at Amazon fit into the already existing culture of the company. The implementation of daily management and kaizen activities at Sonae MC, a leading company that owns more than 300 hyper- and supermarkets in Portugal, was reported by Imai [33]. Based on another three articles Myerson [24] summarizes the adoption of the Lean approach at Starbucks to excel customer experience. The author illustrates the application of a set of Lean practices, including visual management solutions, as well as actions to improve and standardize operational processes. Robinson [100] describes five ways that Zara used Lean to achieve competitive advantages over other fashion retailers, including the adoption of kanban systems to accomplish pulled workflows in their processes.

There are also a number of case studies in the literature reporting the development of improvement initiatives in the retail sector that make use of Lean principles, methods and tools. Jaca et al. [90] describe a Lean project, conducted at a distribution center, that enhanced the productivity rate. In a study conducted by Domingo [101] in a South African retailer, the author determined that about 70% of out-of-stock situations were caused by the stores themselves while only 30% was driven by operational inefficiencies outside the stores. Noda [23] describes aspects of an operational and business Lean transformation that occurred in a mid-size Japanese retailer that sold foods, consumables, apparels, and general merchandise goods. The authors also describe how the company adopted standardized work and process improvement practices based on Lean principles. Evans and Lindsay [102] refer to a Kaizen event conducted at the retail services of Magnivision to investigate the causes of problems that continually plagued employees. Özkavukcu and Durmuşoğlu [103] illustrate how hoshin kanri, a Lean method for strategic planning and deployment can be applied at Migros Ticaret A.Ş, a Turkish food retailer. Eklund [104] reported a study involving nine stores in Sweden where Lean methods were applied to reduce food waste in the fresh food markets. More recently, Abdelhadi [105] conducted research to study how Lean methods and tools can be utilized to prevent the spread of SARS-CoV-2 in a retail store.

## 3. Case Study

The case study was conducted in one store of a multinational retailer located near Lisbon, in Portugal, in 2018. The data date back to 2018, so the research results were not influenced by COVID-19. In addition to the physical store, the hypermarket uses an E-Commerce warehouse to supply most of the Lisbon metropolitan area. In this warehouse

600 best-selling, fast-moving consumer goods product references are stored and some high-volume items that are picked up when a customer places an online order. The remaining items ordered online by a customer are picked up at the store, including items from the fresh food section. This case study aimed at increasing the "order fulfilment rate" accomplished by the online commerce business. This key performance indicator (KPI) is a metric that measures the service level provided to the customer.

### 3.1. Problem Statement

The performance regarding the order fulfilment rate was considered to be poor (<93%). The results for this indicator show a deterioration in performance compared to the previous year and the outlined objective of 95% was far from being achieved. To better understand the factors that most contributed to a change in the value of the order fulfilment rate, the project team developed the construction of a KPI tree, exhibited in Figure 4.

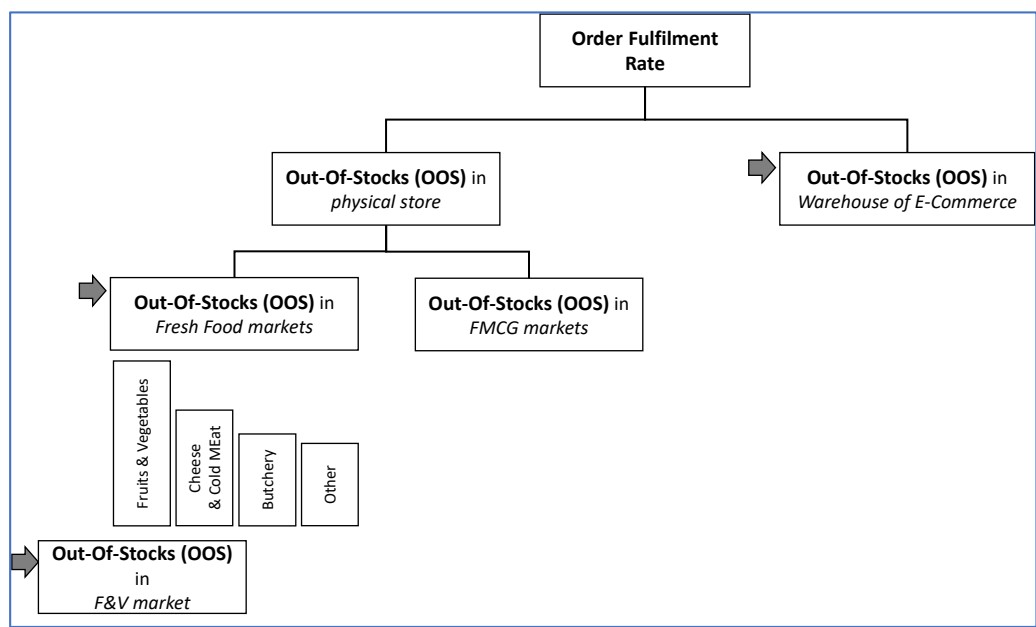

**Figure 4.** KPI tree used to determine the scope and boundaries of the Lean project (own study).

This shows that, to increase the order fulfilment rate, it is necessary to reduce the number of out-of-stocks in the physical store, as well as in the warehouse belonging to the E-Commerce division. Given that the warehouse stores the main references of FMCG products, the team decided to concentrate efforts on the fresh products division. Within this division, the fruits and vegetables (FV) market was the one with the highest incidence of stockouts. The occurrence of stockouts, in its turn, depends on how effective the replenishment processes are to ensure an adequate on-shelf availability. Given the above, the team decided to focus improvement efforts on reducing the number of OOS in two areas:

1. The warehouse of E-Commerce.
2. The market of F V in the physical store.

### 3.2. Methodology

The methodology is depicted in Figure 5, being the branch set for the FV market aligned with the predominant VSM approach proposed by the relevant literature, as described in Section 2.1.1, while the second branch that regards the implementation of kanban cards in the E-Commerce warehouse was specifically proposed under this research. It captures not only the sequence of stages that were followed during both initiatives, but also the Lean approach or tool that was adopted. To reduce OOS in the warehouse it was

decided to introduce a simple and visual inventory management procedure, using kanban cards, for an easy identification of the ordering points for each item as well as the quantity that should be ordered. Regarding the decrease in the number of OOS in the FV section, it was decided to follow the VSM methodology, since it was necessary to understand the existing replenishment process by mapping it and detecting non-value-added tasks and situations that were delaying the expected outcome from the customer viewpoint. The next sections describe in further detail the work carried out during each stage of the methodology for both improvement areas.

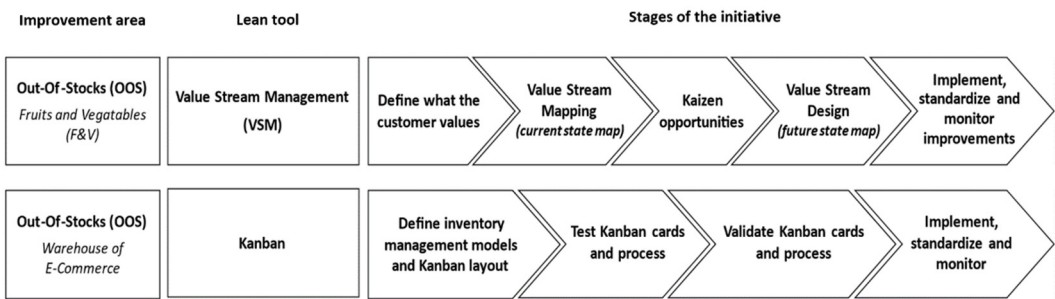

**Figure 5.** Methodology adopted in the case study (own study).

### 3.3. Value Stream Mapping

#### 3.3.1. Stage 1—Define What the Customer Values

The value stream to be improved encompasses all the necessary processes to receive the pallets with the products and check them, dismantle the pallets, restock the items in the correct sales point, perform the FIFO (first-in/first-out) procedure when restocking, replace the items on the sales plate, and remove any damaged product unit from the sales point. There are two customers for the output of this value stream:

1. The consumer that ordered online.
2. The client that buys in the physical store.

The store opens at 9 a.m.; therefore, at that time all the products need to be already replenished, so the consumer who is present in the store can purchase them. The customer who has placed the order is not at the store but is represented by a picker who is already in the store picking up products before the store opens. Table 1 contains the needs of these two types of customers regarding the value stream for replenishing FV products.

**Table 1.** Requirements for the output of the value stream "replenishment of FV products".

| | Customer of the Physical Store | Customer That Ordered Online |
|---|---|---|
| Delivery requirements | Have the desired product available in the correct place from the moment the store is opened | Have the desired product available in the correct place when the picking is needed |
| Quality requirements | Have the product looking fresh and within the expiration date | |

Based on the value proposition from the customer's viewpoint, it was easier to formulate the conditions for any process activity to be labelled as "value-added": any action that directly contributes to taking the product to the point of sale, in the right quantity and along the shortest possible path.

The specific KPIs to be impacted by the improvement of the value stream are those depicted in the left branch of the KPI tree from Figure 4, hence the OOS in the FV market located in the physical store. Ultimately, the product availability, measured by the OOS metric, will impact the online order fulfilment rate.

### 3.3.2. Stage 2—Value Stream Mapping (Current State Map)

This step required a deep understanding of the flows of products, people (including pickers from the E-Commerce service as well as stockers), and information involved in the value stream. It was necessary not only to involve people from different functional areas, but also to perform various "Go and Observe" moments to follow the existing processes along the value stream. To be able to start chartering the current state map, the team felt the necessity to define, or to adapt, some specific VSM symbols and icons, because not all of the standard symbology available in the literature fully fits the nature of the in-store retail operational processes. The proposal of specific VSM icons for retailing operations is one of the contributions of this paper. The result is exhibited on Figure 6.

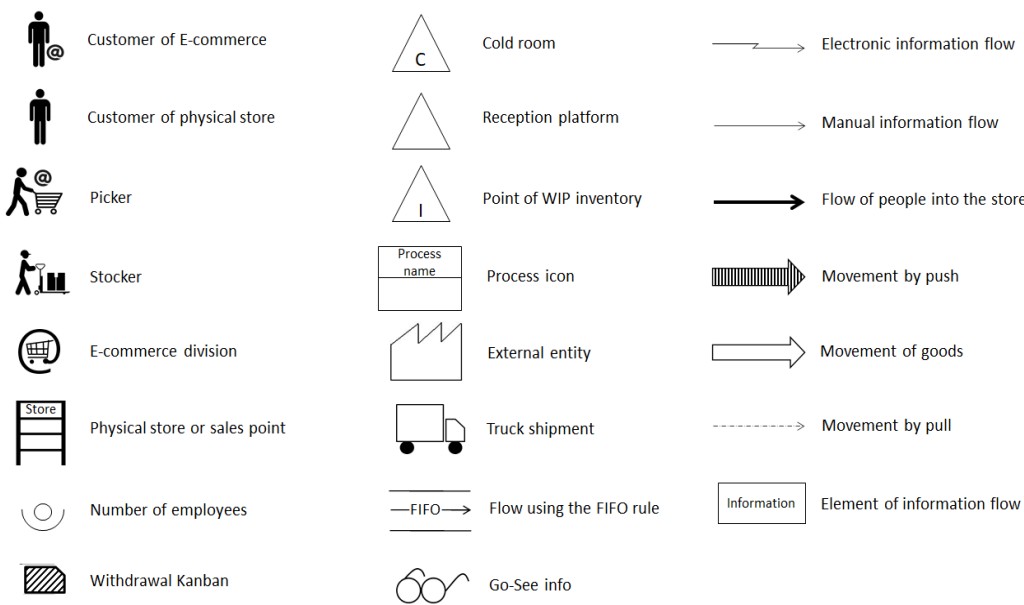

**Figure 6.** Symbols and icons proposed and adopted to construct the value stream maps involving in-store retail processes (own study).

The resulting value stream map of the current sequence of processes is illustrated in Figure 7. The truck bringing the products from the logistics platforms arrives daily in the reception of the store between 4 and 5 a.m., remaining there until the stockers of the FV market pull the pallets—or some of the boxes therein contained—to the store to be replenished. The two stockers usually scheduled for the work enter at 6 a.m. and their first task is to remove from the cold room the products that had been collected at night when the store closed. Based on the FIFO procedure, these will be the first items to be replenished. The pickers from the E-Commerce division start to appear in the FV section by 6:30 a.m. but this is too early, because the refilling of the shelves with products has barely started. Products in this market should be totally restocked by 8:45 a.m., but replenishment usually continues even after the store opens at 9 a.m. This creates issues in terms of product availability for both types of customers. After the pickers collect the required products from the store, each order is shipped to the customer in one of the seven delivery slots. The availability of product in the early morning will significantly affect the first four delivery slots, hence the value of the order fulfilment rate.

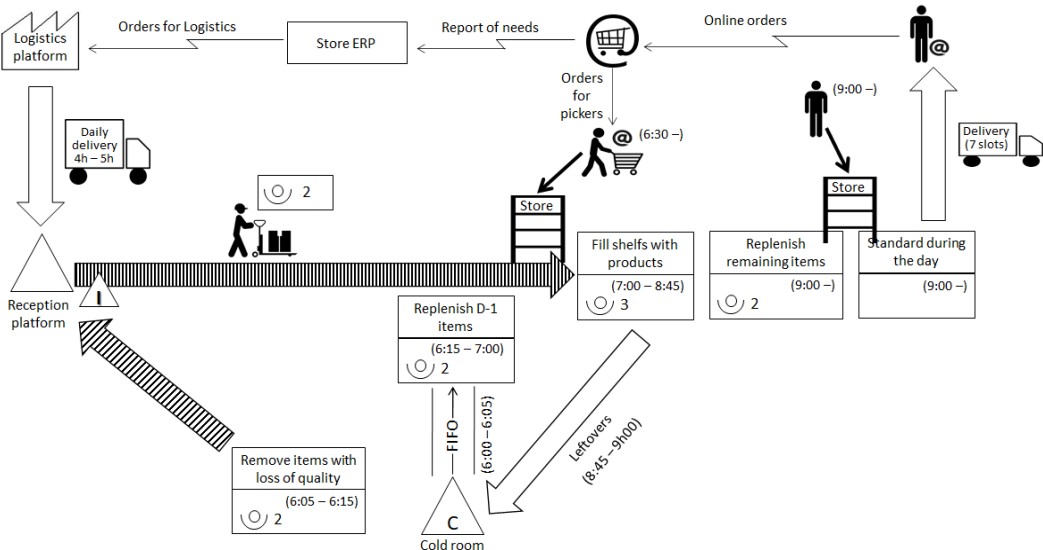

**Figure 7.** Current state value stream map (own study).

The information flow is mainly depicted at the top of the value stream map, and its direction is generally right to left. When the customer places an online order, it is received and treated by the administrative area of E-Commerce, which will then create lists of orders to be given to the pickers. For products that are not stored in their warehouse, which is the case for all FV products, the E-Commerce division needs to inform the store about their needs, attending to the expected online demand. The store is responsible for sending the orders to the central logistics.

This diagnosis allowed the project team to determine the waste factors in the process that could negatively impact the OOS metrics indicated in the previous subsection, hence the online order fulfilment rate.

### 3.3.3. Stage 3—Kaizen Opportunities

The previous stage enabled the team to understand the areas for improvement of the processes under the value stream. The kaizen/improvement opportunities that will impact the KPIs of interest are summarized in Figure 8.

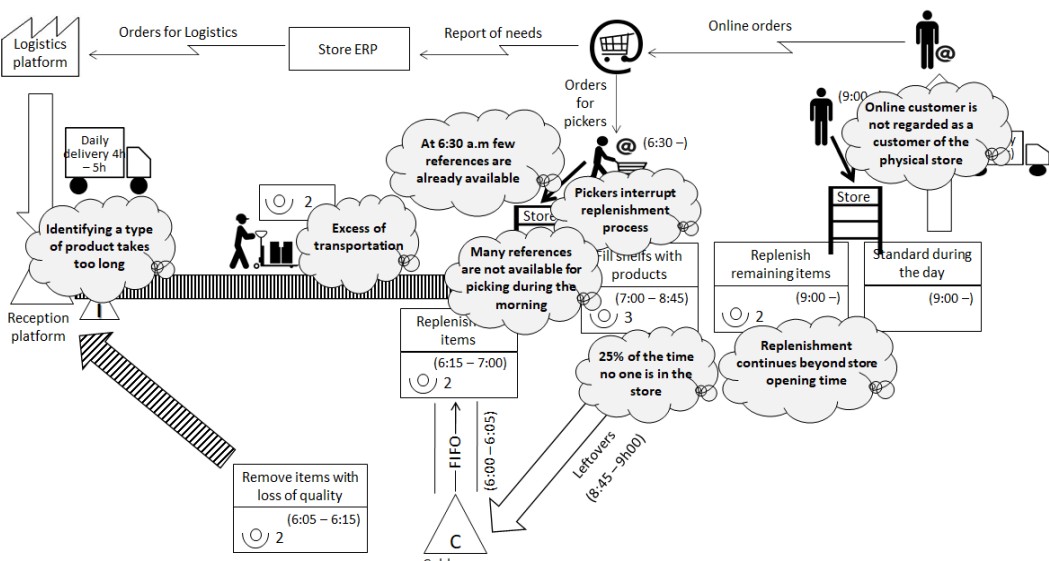

**Figure 8.** Opportunities for improvement identified in the current value stream map (own study).

### 3.3.4. Stage 4—Value Stream Design (Future State Map)

Considering the opportunities for improvement, the team developed a new value stream map regarding the intended future state for it. The resulting map is depicted in Figure 9. The main changes summarized below attempted to introduce a pull flow in the processes, eliminate waiting times, rework, unnecessary movements and in-store travels:

- Instead of remaining in the reception, the pallets are moved by the personnel of this area into the corridor of the store nearby the FV market area. A stocker from the FV market arrives 1 h earlier at 5 a.m. to dismantle the pallets and sort the products by category.
- The other two stockers of the FV market arrive at 6 a.m. to start refilling the shelves. Unlike what happened before, these two people remain in the store and do not need to move to the backroom of the store nor to the reception area. Any necessary transportation or movement is exclusively performed by the worker that arrived earlier, who also has the role of supplying all the goods to be replenished and to remove pallets and empty boxes. Lean calls to this role "Mizusumashi" [106].
- To leverage the workload among work shift, it was decided that the evening shift (responsible for closing the store) would advance the replenishment process by refilling all the products that do not need to remain at a controlled temperature. This takes away needed work time for the tasks necessary to prepare the store opening.
- The managers of the store and of the E-Commerce division agreed that the pickers would only go to the FV section from 7:30 onwards. Between 6 a.m. and 7:30 a.m. the priority for the stockers is to refill all the product references required by the customer that ordered online. Every day, the E-Commerce division communicates to the store the list of products to be picked, so this is similar to a withdrawal kanban system. It avoids unnecessary interruption in the work performed by both the pickers and stockers.
- The pickers will have access to the list of goods that are not available in the store because they were not delivered by the truck that arrived from the logistics platform. This will avoid pickers wasting further time.

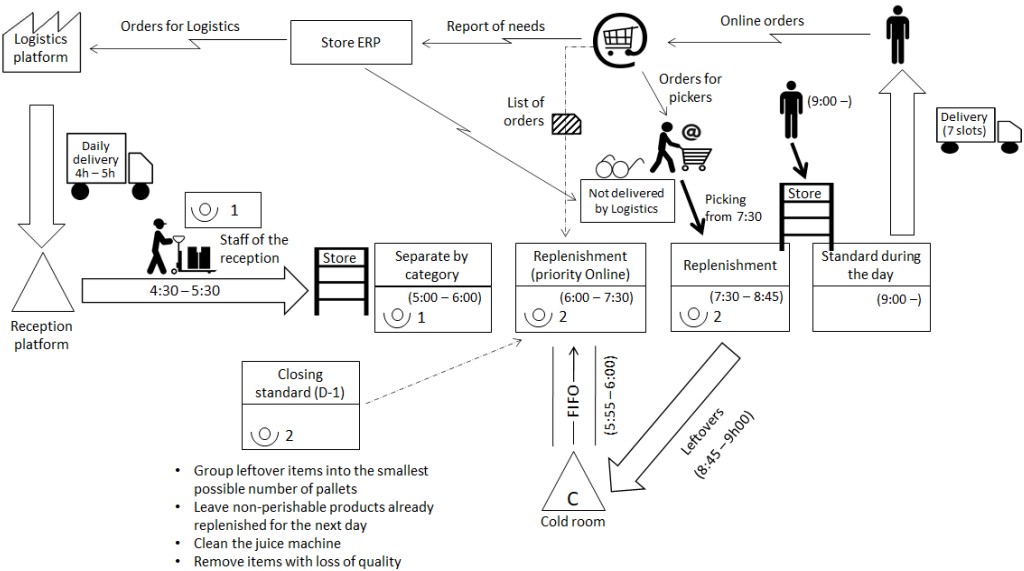

**Figure 9.** Designed future state for the value stream (own study).

Table 2 states the baseline value as well as the objectives set for the KPIs of interest to be impacted by the improvement of the replenishment process in the FV market.

**Table 2.** Baseline and objectives for the KPIs impacted by the replenishment process in the FV market.

| Name of KPI | Baseline | Objective |
|---|---|---|
| Out-of-stock (OOS) rate in the fresh food markets | 6.5% | 5.0% |
| Out-of-stock (OOS) rate in FV | 10.0% | 7.5% |

### 3.3.5. Stage 5—Implement, Standardize and Monitor Improvements

An action plan with the mentioned improvement activities was defined, planned, and implemented in the following weeks. A set of procedures were standardized and the people involved in the processes trained in these standards. The KPIs were followed regularly according to the control plan.

### *3.4. Kanban*

### 3.4.1. Stage 1—Define Inventory Management Models and Kanban Layout

A single card type of kanban was considered adequate to accomplish the objective of implementing a visual inventory management procedure at the E-Commerce warehouse. The purpose of using this tool is to decrease the likelihood of stockouts in this warehouse, measured by the OOS rate for this area. Two important parameters to include in the kanban card were defined: (1) reorder point, and (2) order quantity. The algorithms to determine the values of both parameters were suggested by the supply chain division. The team decided that the cards should include the following information:

- Conversion Factor (number of product items contained in a supplied box of product);
- Photo and description of the product;
- Product European Article Number (EAN) code—a type of barcode that encodes an article number;
- Name of the supplier;
- Days of the week to place an order;
- Days of the week provided for receiving orders.

The kanban cards were defined so that they would be located in the racks of the warehouse near to the product reference they correspond to.

### 3.4.2. Stage 2—Test the Kanban Cards and Process

A few different versions for the layout of the kanban card were proposed, created and tested. The tests started on a specific rack. This allowed the personnel to understand and practice the new procedure: when the ordering point is reached, the worker just needs to scan the EAN code to initiate the ordering process, digitizing the number of units to be ordered.

### 3.4.3. Stage 3—Validate the Kanban Cards and Process

After being tested in a specific rack, the concept was extended to the entire sweet grocery market, and finally generalized to the entire racks of the warehouse. The final validated version of the kanban is visible in Figure 10.

### 3.4.4. Stage 4—Implement, Standardize and Monitor

Similar to the VSM initiative, the new procedures were standardized and a control plan to follow the KPI of interest (OOS in the E-Commerce warehouse) was defined. Table 3 provides the baseline and the objective set for this KPI.

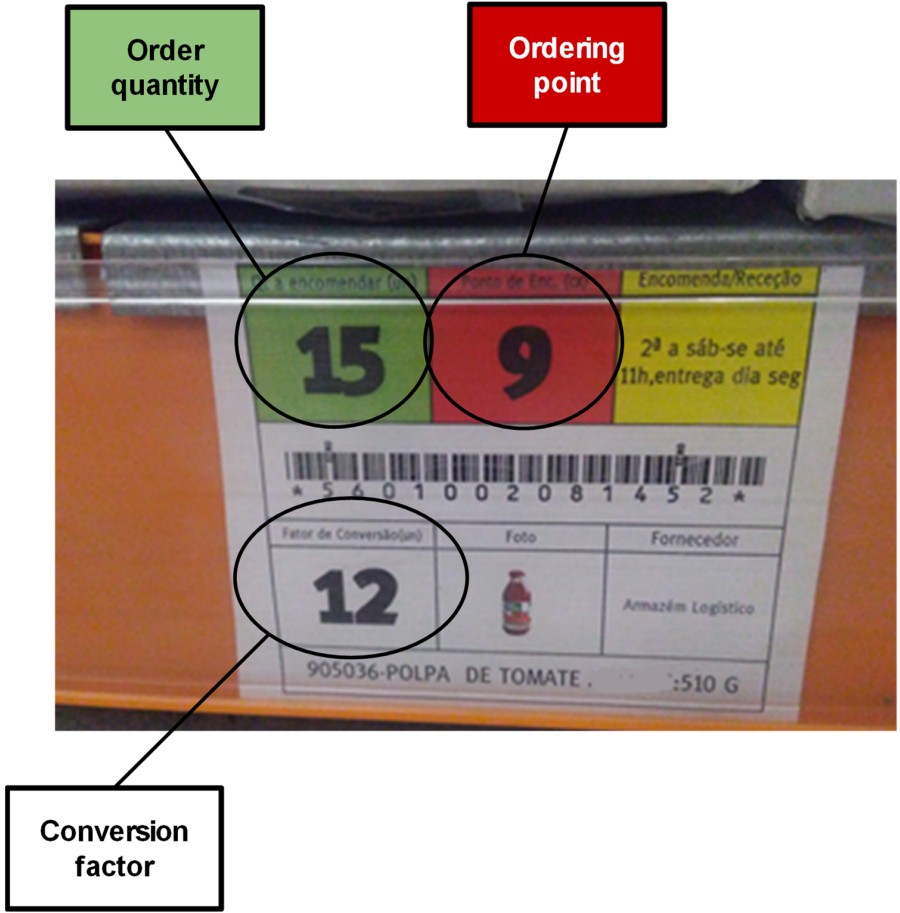

**Figure 10.** Final version of the kanban card used by the E-Commerce warehouse.

**Table 3.** Baseline and objectives for the KPI impacted by the inventory control system in place in the E-Commerce warehouse.

| Name of KPI | Baseline | Objective |
|---|---|---|
| Out-of-stock (OOS) rate in the E-Commerce warehouse | 1.50% | 0.50% |

### 3.5. Research Results

The initiatives described in the previous subsection contributed to an improvement in all the leading and lagging indicators of interest, summarized in Table 4. This table provides the values for these KPIs before and after the improvement initiatives were finalized. This is the first published research that studies the impact Lean process improvement, using the VSM and kanban approaches, on key retail operational indicators that comprise the physical store and E-Commerce businesses.

**Table 4.** Values for the KPIs of interest before and after the improvement initiatives.

| Name of KPI | Before the Initiative | After the Initiative |
|---|---|---|
| Overall Order Fulfilment Rate (overall) | 92.5% | 94.5% |
| Order Fulfilment Rate in F&V | 90.0% | 93.0% |
| OOS in the fresh food markets | 6.5% | 5.8% |
| OOS rate in F&V | 10.0% | 4.2% |
| OOS rate in the E-Commerce warehouse | 1.50% | 0.30% |
| Number Out-Of-Stocks per day (F&V) | 70 stockouts/day | 45 stockouts/day |
| Number Out-Of-Stocks per day (Warehouse) | 10 stockouts/day | 2 stockouts/day |

The results achieved were very positive, showing how Lean methods and tools can be applied to improve the operational performance in a retail environment. In the E-Commerce warehouse, the OOS rate fell to approximately 0.30% (better than the objective set), which represents an evolution of the average number of daily stockouts that decreased from 10 to 2, hence revealing the effectiveness of the kanban system. The improvement of the replenishment process practiced in the FV market also led to a significant enhancement of the achieved performance. It contributed to a reduction of almost 33% of the necessary time to refill the shelves in this market every day, driving a boost in performance of the OOS rate from about 10.0% to 4.2%. In terms of OOS rate performance in the overall fresh food markets, the contribution from the FV market allowed a reduction from 6.5% to 5.8%, the two initiatives made possible an increase of the overall order fulfillment rate: from 90% to 93% overall and from 92.5% to 94.5% in the FV market. The stated objectives were not yet reached, but the performance levels were significantly improved.

The replenishment process for the FV market was adopted by some other stores of the company, but not by the majority of them. Standardizing procedures between stores is usually very difficult, since each one has a certain degree of autonomy; nevertheless, the kanban card solution previously described would be later deployed to a store in the north of Portugal with a smaller E-Commerce warehouse storing about 250 references of FMCG products. That store achieved a reduction in the daily stockouts from 8 to 3 in that warehouse.

### 3.6. Discussion

The contributions of this article are twofold. First, this is the first published case study where Lean was applied simultaneously to improve existing processes in a physical store as well as E-Commerce business processes. Thus, the article stresses that kanban cards allowed a reduction of OOS in the warehouse, through an easy identification of the order points of each item, as well as the quantity to be ordered. On the other hand, the VSM made it possible to reduce the number of OOS in the FV section, since it was necessary to understand the existing replenishment process through its mapping and to detect tasks and situations without added value. As the synergies between these two types of tools are somewhat underexplored in the literature, this article contributes to the literature insofar as it corroborates the arguments that Lean tools and techniques can be effectively adopted by any type of organization to eliminate anything that does not create value [80–82].

Secondly, the managerial contribution is relevant, as it shows that the use of a tool such as VSM requires a deep knowledge of its use. In other words, it is not enough for managers and industrial engineers to have knowledge of the tools. It is necessary to involve different people, both from functional areas and from different levels of the organization. So that they can follow the existing processes along the value stream process. Furthermore, the VSM process is also not static either, as in some circumstances it is necessary to define, or adapt, some specific symbols and icons. Since not all the standard symbology available in the literature fully suits the nature of the organization.

### 4. Conclusions

Due to fast technological changes, intense competition, and the emergence of new business models, among other factors, many retailers have sought to incorporate productivity and efficiency improvement practices into their processes through the development and implementation of Lean programs. This paper presents a case study conducted in a Portuguese retail hypermarket, which aimed to demonstrate that Lean concepts, methods and tools can be successfully applied in a company from the retail industry.

The project herein described encompasses two initiatives that together sought to increase the accomplished E-Commerce order fulfilment rate. The performance of this indicator depends on the OOS value in the store as well as in the existing warehouse in the backroom of the store managed by the E-Commerce division. The OOS reduction in the warehouse was achieved through the implementation of a simple inventory management

procedure using a visual kanban card, one that demonstrated the usefulness of this Lean tool. The warehouse stores the 600 best-selling product references belonging to the FMCG category; for this reason, efforts to reduce the number of OOS on the store itself did not focus on this type of product, but instead fell on the most relevant of the fresh food markets: FV. In this case, the value stream management approach was adopted to design and implement a more efficient and faster replenishment process. Both initiatives resulted in a significant decrease of the stockouts, driving an important improvement of the order fulfilment rate.

**5. Limitations and Future Work**

To the best knowledge of the authors, this is the first published case study where Lean was simultaneously applied to improve existing processes in a physical store as well as processes from the online commerce business. Some of the synergies between both kinds of processes were explored. A limitation of this study is the fact that it was only conducted in one store. In addition, the research was conducted in a hypermarket, so it would be interesting in the future to widen the study to other store formats. Regarding directions for future research in this field, some ideas are presented. First, it will be relevant to extend the application of the described methodology to the other food markets, in addition to FV. Second, the study should be conducted in other stores and with different format sizes. A third proposal for future research is related to the practical application of a Lean Six Sigma project in the retail sector, since the available literature is very scarce regarding this topic.

**Author Contributions:** Conceptualization, P.A.M., D.J. and J.R.; methodology, P.A.M., D.J. and J.R.; formal analysis, P.A.M., D.J. and J.R.; investigation, P.A.M., D.J. and J.R.; resources, P.A.M., D.J. and J.R.; writing—original draft preparation, P.A.M., D.J. and J.R.; writing—review and editing, P.A.M., D.J. and J.R.; project administration, P.A.M., D.J. and J.R.; funding acquisition, P.A.M., D.J. and J.R. All authors have read and agreed to the published version of the manuscript.

**Funding:** This research received no external funding.

**Institutional Review Board Statement:** Not applicable.

**Informed Consent Statement:** Not applicable.

**Data Availability Statement:** Restrictions apply to the availability of the data.

**Conflicts of Interest:** The authors declare no conflict of interest.

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
