# Peer review of "Using Lean to Improve Operational Performance in a Retail Store and E-Commerce Service: A Portuguese Case Study"

_sustainability, doi:10.3390/su14105913_

Round 1

Reviewer 1 Report

Dear Authors

Thank you for an interesting paper. I read it carefully. I was very curious about your point of view in case of Lean and e-commerce. I could see that the presentation of the results but also use of Lean assumption and VSM is conducted in proper way.

Your paper is well constructed, includes all necessary section. In abstract, all content is well summarized. The introduction explains the reason why authors decided to write about this subject. Methodology and results are presented in reasonable way. All paper seems to be good written and reasonable.

However, I have some suggestions which can make your paper more valuable. Some of them you can treat only as suggestions (I will mark which one).

  1. Abstract: I would suggest to add 2-3 sentences about main conclusion and your contribution to science.
  2. Key words: be sure that 2-3 keywords are those from SI proposal.
  3. Introduction: emphasize your contribution to science, what is new, what is valuable for others. I would also suggest to underline the connection with sustainability.
  4. Line 136, “Liker ]79]”. Correct for [79].
  5. Figures: if any of the figure is yours, please add this info, for example “own study”. I am not sure about in case of couple of your figures. Figure 1, is the title correct? Figure 6, these symbols are well known, so I think source in this case is really needed.
  6. Point 3.3.4. You have bullet point. I think at the end your should use a dot, not semicolon.
  7. Point 3.5. There is lack of study results discussion in relation to previously published results. I would suggest to add something.
  8. Point 4. Study limitations and future study discussion are rather weak.
  9. And here I would like to ask you to work a bit. The list of references is too long list. When introducing the theory you used too many sources in one time, like you wanted to do it on purpose. I am sure many of them are necessary, I suggest to leave 50%=60% of the most important literature.

Beside all my comment I think the paper is suitable to be published, of course after all correction. You did a great job.

Author Response

Dear reviewer.

Thank you for your contribution.

Reviewer 2 Report

  • In all the text there are many abbreviations used by authors, please kept them in minimum, but if they are known in their literature that’s ok, just first they must be written in full phrase in small letter (most of them written in capital form like; Motor Vehicle Program (IMVP), instead use them in small letter (motor vehicle program (IMVP), if it is in first of sentence then must be in capital (Motor vehicle program (IMVP). 
  • In section 2.1.1 you already used abbreviation for VSM, again you wrote full phrase in continue please correct it. Again, in section 3.2. please check VSM. Check all acronyms again. 
  • Before conclusions, managerial implications of work is missing. Please strength this section. 

Author Response

(The authors gave the same response as above.)

Reviewer 3 Report

Dear Authors,

The paper presents a relevant topic and addresses relevant aspects to be analyzed and improved. However, there are items that are recommended to be improved before considering the paper for publication:

  1. The abstract needs to specify better in which areas does the paper provides a novelty and a contribution of the current state of the art.
  2. The practical need and challenge is clear stated however, the introduction to the topic needs to describe the challenge and research gaps from the point of view of research more in detail, thus, the introduction section is required to provide more evidences and references of previous studies in this area to derive why the presented research contributes and based on which elements as for instance e-commerce with lean and stock monitoring
  3. The title of figure 1 seems to be not appropriate. Moreover, this figure seems also to have some relation to some literature review and the content is quite small to be read properly.
  4. Chapter 2 needs to be shortened and focused to the subject of the research.
  5. Research process: the paper presents a methodology in Figure 5, however it is unclear which part of the methodology has been derived or extracted from literature and which ones have been developed within the research study. Moreover, it should be defined more clearly how the steps have been determined.
  6. In Stages 3, 4 and 5 of the VSM, more details are needed with specific measures, development and implementation milestones as well as expected and obtained results (qualitative or quantitative).
  7. Idem as point 6 for stages 2,3 and 4 for Kanban.
  8. A recommendation would be to separate results and discussion. Results with more detail definition of KPIs and a discussion section in which the paper compares the findings with other existing methods and research studies in the field.
  9. The conclusions should be more linked to chapter 3.
  10. The sentence “To the best knowledge of the authors, this is the first published case study where Lean was simultaneously applied to improve existing processes in a physical store as well as processes from the online commerce business” should be checked with existing research and studies. If this is a main contribution of the paper it should be better highlighted during the paper including the results.
  11. There is a need to include limitations and future work.

Best regards

Author Response

(The authors gave the same response as above.)

Round 2

Reviewer 3 Report

The paper can be published in its present form